# Developing a Multi-Spectral NIR LED-Based Instrument for the Detection of Pesticide Residues Containing Chlorpyrifos-Methyl in Rough, Brown, and Milled Rice

**DOI:** 10.3390/s24134055

**Published:** 2024-06-21

**Authors:** Fatima Rodriguez-Macadaeg, Paul R. Armstrong, Elizabeth B. Maghirang, Erin D. Scully, Daniel L. Brabec, Frank H. Arthur, Arlene D. Adviento-Borbe, Kevin F. Yaptenco, Delfin C. Suministrado

**Affiliations:** 1Faculty of Institute of Agricultural and Biosystems Engineering, Don Mariano Marcos Memorial State University-North La Union Campus, Bacnotan 2515, Philippines; 2CGAHR—Center for Grain and Animal Health Research, ARS-USDA—Agricultural Research Service, Manhattan, KS 66502, USA; paul.armstrong1956@gmail.com (P.R.A.); elboma@yahoo.com (E.B.M.); erin.scully@usda.gov (E.D.S.); dan.brabec@usda.gov (D.L.B.); frank.arthur@usda.gov (F.H.A.); 3DWMRU—Delta Water Management Research, ARS-USDA—Agricultural Research Service, Jonesboro, AR 72467, USA; arlene.advientoborbe@usda.gov; 4Faculty of the Institute of Agricutlural and Biosystems Engineering, University of the Philippines Los Baños, Laguna 4031, Philippines; kfyaptenco@up.edu.ph (K.F.Y.); dcsuministrado@yahoo.com (D.C.S.)

**Keywords:** chlorpyrifos-methyl pesticide residue (CMPR), light emitting diode, near infrared, organophosphates, pesticide residue, rice

## Abstract

A recent study showed the potential of the DA Perten 7200 NIR Spectrometer in detecting chlorpyrifos-methyl pesticide residue in rough, brown, and milled rice. However, this instrument is still lab-based and generally suited for point-of-sale testing. To provide a field-deployable version of this technique, an existing light emitting diode (LED)-based instrument that provides discrete NIR wavelength illumination and reflectance spectra over the range of 850–1550 nm was tested. Spectra were collected from rough, brown, and milled rice at different pesticide concentrations and analyzed for quantitative and qualitative measurement using partial least squares regression (PLS) and discriminant analysis (DA). Simulations for two LED-based instruments were also evaluated using corresponding segments of spectra from the DA7200 to represent LED illumination. For the simulation of the existing LED-based instrument (LEDPrototype1) fitted with 850, 910, 940, 970, 1070, 1200, 1300, 1450, and 1550 nm LED wavelengths, resulting R^2^ ranged from 0.52 to 0.71, and the correct classification was 70.4% to 100%. The simulation of a second LED instrument (LEDPrototype2) fitted with 980, 1050, 1200, 1300, 1450, 1550, 1600, and 1650 nm LED wavelengths showed R^2^ of 0.59 to 0.82 and correct classifications of 66% to 100%. These LED wavelengths were selected based on the significant wavelength regions from the PLS regression coefficients of DA7200 and the commercial availability of LED wavelengths. Results showed that it is possible to use a multi-spectral LED-based instrument to detect varying levels of chlorpyrifos-methyl pesticide residue in rough, brown, and milled rice.

## 1. Introduction

To ensure the proper handling and safe consumption of rice, the Food and Agriculture Organization (FAO) and the World Health Organization (WHO) have imposed maximum residue limits (MRLs) for toxic pesticides [1], including chlorpyrifos-methyl. If used inappropriately and excessively, this pesticide has been reported to have toxic effects on the human body, particularly on the brain and nervous system [2,3]. Based on independent epidemiological, in vivo, and in vitro studies, the evidence points to adverse health effects from exposure to chlorpyrifos on the developing nervous system, which has been associated with lowered IQ at school age [3].

Rice is the main staple food in the Philippines. To ensure ample supply, importation from countries such as Thailand and Vietnam has been a common practice. In 2018, a shipment of 330,000 bags of 50 kg milled rice from these two countries was found to be infested by rice weevils, and samples was then treated with formalin-laced insecticide by the National Food Authority (NFA) [4]. This highlighted the need to detect pesticide residues as consumers have expressed concern for the potential risk to human health, especially when pesticide residues remain undetermined during handling and before human consumption. However, the commonly used detection technique for the presence of pesticide residues in grains has been gas and liquid chromatography, which involves specialized instrumentation and labor-intensive protocol for extraction, centrifugation, cleanup, evaporation, and scan of rice samples [5]. The development of a technique or simple low-cost instrument with a fast turnaround in the detection of pesticide residues will highly benefit consumers, safety regulators, and the rice industry.

Near-infrared spectroscopy (NIRS) is a rapid, precise, and non-destructive technique that has shown potential in the determination of numerous chemical and physical properties of foods and food products [6]. An earlier study conducted by Rodriguez et al. [7] showed the potential of using NIRS for detecting pesticide residues that contain varying concentrations of chlorpyrifos-methyl in rough, brown, and milled rice using a commercial full wavelength (950–1650 nm) NIR instrument (Perten DA7200, Perten Industries, Springfield, IL, USA). Chlorpyrifos-methyl pesticide was selected as the target pesticide residue for investigation in this study because of the widespread use of chlorpyrifos for the control of insect infestation in the Philippines and the underlying health concerns related to this pesticide. While the NIR spectroscopy technique is simple, and the commercially available NIR instruments can be readily adapted, there is a need in countries such as the Philippines for a fast and reliable technique or instrument that is low-cost and made of locally available parts. Portability is also preferable, as it allows for testing at various handling and pre-consumption points.

An existing tabletop LED-based prototype instrument (LEDPrototype1) being developed at the U.S. Department of Agriculture (USDA), Agricultural Research Service (ARS), Center for Grain and Animal Health Research in Manhattan, Kansas, which was based on the design of the USDA-ARS multi-spectral high-speed sorter [8], provided a potential platform for an instrument to be used for the detection of pesticide residues. The prototype, referred to as LEDPrototype1, was designed for bulk samples and uses an array of NIR LEDs, an NIR detector, and a microprocessor. Although the wavelengths of the LEDs in LEDPrototype1 were not the same combination as what was found to be important in earlier studies by Rodriguez et al. [7] using a commercial spectrometer (DA7200) [9], the evaluation of the prototype was considered important to determine the potential contributions from other wavelengths and also to verify the light, signal, and noise filtering capabilities of LEDPrototype1.

The main objectives of this study were to (1) simulate the performance of a proposed multi-spectral NIR LED-based instrument that uses the wavelengths that were identified to be important for the detection of varying concentrations of chlorpyrifos-methyl pesticide residues (CMPR) in various rice types (rough, brown, and milled rice) and (2) evaluate the existing LEDPrototype1 to obtain relevant design considerations such as light, signal, and noise filtering capabilities. Findings from this study will be used for future work on the design and development of a multi-spectral NIR LED-based instrument that can be fabricated using low-cost and available parts in the Philippines. Considering the versatility of NIR spectroscopy at measuring and detecting different chemical compositions, it is also possible that the instrument being developed can be used for other applications such as the measurement of moisture, protein, oil, and starch contents for different grain and food products.

## 2. Materials and Methods

### 2.1. Rice Grain Samples

The samples of rough, brown, and milled rice used in this study were the same samples used in an earlier study by Rodriguez et al. [7]. Briefly, five pesticide-free rough rice varieties (CL151, Diamond, Hybrid 1, Gemini, and Hybrid 2) were provided to USDA-ARS, Manhattan, KS, USA by the Delta Water Management Research Unit of USDA-ARS, Harrisburg, Arkansas. All rough rice samples were cleaned using the Carter Day dockage tester (Carter Day International, Minneapolis, MN, USA) and conditioned to 14% using an environmental chamber (Percival Intellus Control System, Percival Scientific, Fontana, WI, USA). The chamber was set at 23 °C and 73% relative humidity. The rice samples were spread thinly on trays and were conditioned inside the chamber for three days.

Brown rice samples were prepared by hulling using a JLGL-45 rubber roller (Wuhan Acme Agro-Tech Co. Ltd., Wuhan, China). A portion of the brown rice was further polished using a Twinbird MR-E500 mill (Twinbird Corp., Tsubame City, Japan) to obtain milled samples. The samples were divided into 110 g subsamples using a Boerner divider (Sedburo Equipment, Des Plaines, IL, USA) and were placed and sealed in 946 mL clear, wide-mouth plastic jars.

For the application of treatment in rice samples, each of the 90 subsamples (5 varieties at 6 pesticide concentrations replicated 3 times) for each rice type was spread onto a layer of kraft paper and sprayed evenly with 0.2 mL of water for the control samples (no pesticide applied) and 0.2 mL of pesticide concentrations for the rest of the samples that were treated with varying levels of pesticide solution containing chlorpyrifos-methyl. The pesticide solution was prepared by diluting StorcideTM II (21.60% chlorpyrifos-methyl) with water to attain target pesticide concentrations, i.e., (a) 1.5, 3, 6, 9, and 12 ppm for rough rice; (b) 0.75, 1.5, 3, 4.5, and 6 ppm for brown rice; and (c) 0.1, 0.2, 0.4, 0.6, and 0.8 ppm for milled rice samples. The concentrations were based on the maximum residue limit of CMPR in rough, brown, and milled rice [1] and the application rate of chlorpyrifos-methyl [10]. The treated rough and milled rice samples were immediately placed back in the plastic jars and sealed before spectral data collection. The treated brown rice samples were vacuum-sealed using a FoodSaver Vac 360 (Sunbeam Products, Inc., Boca Raton, FL, USA) to minimize possible lipid degradation during storage prior to spectral data collection.

### 2.2. Instrumentation

Two NIR instruments were evaluated for their potential to detect varying concentrations of CMPR in rice (rough, brown, and milled). These instruments were a commercially available Perten DA7200 (Perten Industries, Springfield, IL, USA) and the ARS-USDA-designed LEDPrototype1. Both are designed for bulk analysis.

Based on a previous study conducted by Rodriguez et al. [7], the 950-to-1650 nm wavelength range of the DA 7200 allowed for the detection of CMPR in rough, brown, and milled rice. DA7200 has a wavelength accuracy of <0.3 nm and stability of <0.2 nm/year; it uses a 256-element indium gallium arsenide (InGaAs) diode detector array, which is thermoelectrically cooled. The instrument collects spectral data on samples placed in an open-faced sampling dish in ambient room light at ~100 spectra/s at 3.125 nm/diode resolution with an accuracy of ±0.02 a_w_. It operates using Windows XP platform with 1 Ghz, 256 MB RAM, and 20 GB HDD. The system automatically corrects possible background noise by collecting spectra without samples first, followed by spectra with samples [9]. The LEDPrototype1 instrument (Figure 1) was based on a design used for a multi-spectral high-speed, single seed sorter [8]. The LEDPrototype1 instrument circuit board is composed of LED light sources, an InGaAs detector, a signal amplifier, and a microprocessor. The LEDs in this instrument were selected to cover a broad spectrum range from LED wavelengths that were commercially available. LED wavelengths were 850, 910, 940, 970, 1070, 1200, 1300, 1450, and 1550 nm. LEDs were arranged in a circular pattern around the lens and directed to a central point on the grain surface 12 cm directly above the lens. The LEDs emitted a narrow light beam that was approximately dispersed over ±10° from the center (Figure 2). Data acquisition and LED sequential pulsing were achieved using a microcontroller (ATmega328P Atmel Corp., San Jose, CA, USA). An InGaAs photodiode (SD060-11-41-211, Luna Optoelectronics, Camarillo, CA, USA) with high sensitivity, low noise, and 1 mm diameter active area for spectral detection (800–1700 nm) was used to detect reflected light and was amplified by a trans-impedance amplifier (OPA2380, Texas Instruments, Dallas, TX, USA). The sensor board was placed inside a black enclosure to eliminate ambient light; communication to the laptop for data collection was via USB. A graphical user interface program (GUI) was created such that the instrument can connect to a laptop COM port at a 115,200 baud rate and control some of the acquisition parameters. The Atmel AVR Studio 5.1 (Atmel Corp., San Jose, CA, USA) was used to program the microcontroller to send and receive digital and analog I/O and download data to the laptop. Spectral collection on 100 g samples was carried out by rotating a shallow circular dish containing the sample, 76 mm diameterby 38 mm deep, placed on a small rotary table turning at 6 rpm. The sensor circuit board faced downward toward the sample being scanned.

### 2.3. Spectral Data Acquisition

The two instruments were used to obtain spectra for each of the 90 rice samples. For DA7200 and LEDPrototype1, three replicates were carried out for each sample with three repacks per replicate. The bulk rice sample to be scanned was placed in the sample dish and set in the viewing area of each of the instruments. To address potential issues of cross-contamination of pesticide residues, the sample dishes used for both instruments were washed with soap and water and then immediately dried using compressed air after each spectral collection.

### 2.4. Data Analysis

Quantitative and qualitative spectral data analyses were performed using partial least squares (PLS) regression and discriminant analysis (DA), respectively, using the UnscramblerX version 10.5.1 (CAMO Software Version 10.5.1, Oslo, Norway). For both analyses, the 270 spectral data for each rice type were divided into a calibration (*n* = 216) and independent validation (*n* = 54) sample sets based on the leave-one-variety-out sampling method. This method involved using the spectral data for different pesticide residue concentrations from four of the five varieties as calibration samples, while the spectra from the remaining rice variety were used as the independent validation samples.

Aside from the analysis with the independent validation sample set, an analysis that made use of all varieties with cross-validation was also performed. Several pretreatments, including mean-centering, standard normal variate, multiplicative scatter correction, and derivative techniques, were evaluated. The performance of the calibration models when using the different pretreatments was similar, which was also observed by Rodriguez et al. [7]. Thus, only the simplest models that made use of mean-centering were presented.

For quantitative analysis, the pesticide concentrations for the sample treatments were used as reference data, i.e., 1.5, 3, 6, 9, and 12 ppm for rough rice; 0.75, 1.5, 3, 4.5, and 6 ppm for brown rice; and 0.1, 0.2, 0.4, 0.6, and 0.8 ppm for milled rice samples. For qualitative analysis, the six levels of pesticide concentration were pooled into two groups: a low pesticide level (LPL) and contaminated or high pesticide level (HPL). For rough rice, the LPL group included 3 ppm and below, while the HPL was 6 ppm and higher. For brown rice, the LPL group included 1.5 ppm and below, and HPL was defined as 3 ppm and higher. In milled rice, LPL was 0.2 and below, and HPL was 0.4 ppm and higher. A value of “1” was assigned to LPL and “2” to HPL samples for discriminant analysis.

## 3. Results and Discussion

The wavelengths that contributed most toward the quantitative and qualitative measurements of pesticide residues containing varying concentrations of chlorpyrifos-methyl in rough, brown, and milled rice based on earlier evaluations using DA7200 were provided by Rodriguez et al. [7], as summarized in Table 1. These were identified based on the peaks and valleys of the plotted regression coefficients for the full wavelength (950 to 1650 nm), which represent the wavelengths that contributed most to the prediction capability of the model.

Based on the functional groups present in chlorpyrifos-methyl, potential changes in the composition of the grains, and their respective adsorption wavelengths [8,11], the significant wavelengths identified for rough rice were 980 nm (starch), 1050, 1390, and 1410 nm (oil), 1200 nm (C-H bonds), 1360 nm (methyl), 1425 nm (protein), 1480 nm (N-H bonds), and 1540 nm (amine). Important wavelengths for brown rice were 1200 and 1300 nm (C-H bonds), 1360 nm (methyl), 1450 nm (C=O bonds), 1470 nm (N-H bonds), and 1580 nm (amide), while those for milled rice included 1200 nm (C-H bonds), 1360 nm (methyl), 1410 nm (O-H bonds), 1510 nm (oil), and 1540 nm (amine). Of these wavelengths, three were already being utilized in LEDPrototype1, i.e., 1200, 1300, and 1450 nm. Other LEDs available in LEDPrototype1 included 850, 910, 940, 1070, and 1550 nm.

The wavelengths that were important regardless of the type of rice were considered for inclusion in the selection of wavelengths for LEDPrototype2. The 1200 and 1360 nm wavelengths were found to be important across all rice types. The 1200 nm LED was readily available and was included in the wavelength to be used. The 1360 nm LED, however, was not commercially available. In this situation, the available LED wavelengths that were closest to the desired wavelengths were selected, which in this specific case was 1300 nm. Other examples are the 1540 nm wavelength, which was important for rough and milled rice and was substituted by the commercially available 1550 nm LED, and the 1580 nm wavelength, which was important for brown rice and substituted by 1600 nm and can also be covered by the 1550 nm. Other important wavelengths with readily available LEDs included 1050, 1200, 1300, 1450, and 1550 nm. Following this selection method, the eight LEDs were selected and used for the design of a multi-spectral NIR instrument. Considering that the instrument being designed was also aimed to be used for other commodities and/or detection parameters aside from CMPR in rice, 980 nm was also included. This selection was based on studies that showed that rice properties can be detected at 740–1070 nm [12].

An instrument simulation, referred to as the DA7200 analog, used wavelengths within 80% relative radiant intensity of the central wavelength used in LEDPrototype1 but using DA7200 spectra for developing calibration models from these discrete ranges. Ranges from DA7200 spectra included 1055 to 1085 nm, 1185 to 1215 nm, 1275 to 1320 nm, 1420 to 1470 nm, and 1515 to 1580 nm. The prediction statistics for quantitative analysis (Table 2) show that for independent validation, rough rice had R^2^s = 0.52 to 0.71 and SEPs = 2.28 to 2.95; in brown rice, R^2^s = 0.58 to 0.70 and SEPs = 1.17 to 1.38 were observed, while milled rice models had R^2^s = 0.57 to 0.71 and SEPs = 0.16 to 0.19. These results indicated poor to marginal quantitative prediction of CMPR at best. Qualitative predictions at different thresholds based on rice milling treatment yielded the highest percent correct classification (%CC) for milled rice at 90.7 to 100%CC followed by rough rice, at 77.8 to 92.6%CC, and brown rice at 70.4 to 88.6%CC (Table 3). These provided indications that using the limited wavelength regions in the LEDProtoype1 should provide good qualitative predictions of CMPR in milled rice.

Actual instrument tests were conducted to detect CMPR using the existing design configuration of LEDPrototype1. Table 4 summarizes the PLS model calibration and validation statistics for quantitative determination in rough, brown, and milled rice. Validation results were poor to very poor in all cases. Across the five calibration models, independent validation results for rough rice revealed R^2^s = 0.23 to 0.59, while brown and milled rice were found to have R^2^s ≤ 0.03. According to discriminant analysis, the %CC for the five leave-one-variety-out calibration models showed some potential for rough rice, yielding 71.6 to 85.8%CC. Brown rice and milled rice were poorer in terms of %CC, ranging from 58.0 to 63.6 and 55.6 to 61.1, respectively (Table 5). This may be attributed to the low amount of pesticide applied. It could be that lower concentrations (1.5 ppm and below) of chlorpyrifos-methyl pesticide residue are low input signals for the detector. Likewise, based on the study conducted by Yao et al. [13], low absorption of samples corresponds to the low information content of samples, while high absorption corresponds to low light transmittance and loud noise, which all interfere with spectral modeling analysis. Yao et al. [13] used absorbance value optimization partial least squares (AVO-PLS) in selecting wavelength in this case.

There was a substantial reduction in prediction performance between the LEDPrototype1- DA7200 analog and the actual sample tests using the LEDPrototype1 instrument. For example, milled rice that can potentially be discriminated as containing low versus high CMPR with 90.7 to 100%CC was only at 55.6 to 61.1%CC in the actual test. This may indicate that, while the wavelengths that are needed for effective prediction are available, the capability was not fully utilized, which may highlight the need for improving the instrument design with a focus on improving light signal and noise filtering.

Based on an evaluation of the results from PLS analysis for DA7200 and LEDPrototype1 and working within currently available LEDs, the following LEDs were selected for use in the proposed NIR LEDPrototye2: wavelengths 980, 1050, 1200, 1300, 1450, 1550, 1600, and 1650 nm. NIR region wavelengths of 1360, 1390, 1410, 1425, 1470, 1480, 1510, 1540, and 1580 nm were considered important for residue detection but are not commercially available, although there is some overlap with those used in this study in some cases.

As noted earlier, these wavelengths are not discrete, and absorption bands for water, protein, starch, cellulose, and oil/fat can occur over a range of several nanometers [14,15]. For example, the protein absorption band at 1186 nm and starch band at 1200 nm can potentially be detected using an LED that covers both of these wavelength ranges. Figure 3 provides an overlapped representation of the relative spectral emission for the LEDs selected for the proposed LEDPrototype2, as provided by the manufacturer (Marubeni America Corporation, Santa Clara, CA, USA) [15]. As an example, the 1540 nm wavelength was among the identified important model wavelengths and pertains to amine absorption [14]; amine is a functional group present in chlorpyrifos-methyl. The 1540 nm LED is not commercially available; thus, the 1550 nm LED (1500 to 1600 nm full range; 1515 to 1580 nm at 80% relative radiant intensity) was selected as a substitute. The 1360 nm wavelength, which is relevant to methyl absorption, is likewise a functional group present in chlorpyrifos-methyl but was not represented in the proposed LEDPrototype2 instrument because the closest available LED was 1300 nm (1250 to 1350 nm at full range; 1275 to 1320 nm at 80% relative radiant intensity). As more LED wavelengths become available, important wavelength(s) for specific applications can be added to potentially improve performance and also widen the range of applications. For LEDPrototype2, considering that the emission spectra of the LEDs approximated a Gaussian curve with a full width at half maximum of approximately 50 nm centered about the peak emission wavelength [7], the replacement LED wavelengths selected accounted for most of the wavelengths that were considered important for the detection of CMPR.

A simulation (DA7200 analog) was carried out to determine the performance of the proposed LEDPrototype2. Most of the wavelengths available in the 950–1650 nm DA7200 spectrum are covered in the selected LEDs for NIR LEDPrototype2 at 80% relative radiant intensity, and as such, performances of the two instruments were found to be comparable (Figure 2). The wavelengths used pertain to amines, oil, protein, water, and starch [8,11]. Table 6 and Table 7 provide the PLS analysis (quantitative) and discriminant analysis (qualitative), respectively, for the proposed LEDPrototype2.

**Table 5 sensors-24-04055-t005:** Discriminant model prediction statistics for determination of chlorpyrifos-methyl residues using the actual NIR LED-based Prototype 1 instrument.

Model Data	Calibration (Training Set)			Independent Validation (Test Set)		
	Numb of False Positives	Number of False Negatives	Overall % CC	Number of False Positives	Number of False Negatives	Overall % CC
Rough Rice (Low: ≤3.0 ppm, High: ≥6.0 ppm)						
ALL varieties ^[a]^	126/405	52/405	78.0 (632/810 ^[N]^)	-	-	-
CL151 ^[b]^	107/324	44/324	76.7 (497/648)	21/81	2/81	85.8 (21/162)
Diamond ^[b]^	102/324	42/324	77.8 (504/648)	20/81	8/81	82.7 (28/162)
Hybrid1 ^[b]^	99/324	41/324	78.4 (508/648)	15/81	17/81	80.3 (32/162)
Gemini ^[b]^	87/324	37/324	80.9 (524/648)	27/81	19/81	71.6 (46/162)
Hybrid2 ^[b]^	100/324	27/324	80.4 (521/648)	36/81	9/81	72.2 (45/162)
Brown Rice (Low: ≤1.5 ppm, High: ≥3.0 ppm)						
ALL varieties ^[a]^	153/405	163/405	61.0 (494/810 ^[N]^)	-	-	-
CL151 ^[b]^	126/324	137/324	59.4 (385/648)	6/81	55/81	62.4 (101/162)
Diamond ^[b]^	116/324	138/324	60.8 (394/648)	34/81	25/81	63.6 (103/162)
Hybrid1 ^[b]^	111/324	138/324	61.6 (399/648)	57/81	5/81	61.7 (100/162)
Gemini ^[b]^	118/324	133/324	61.3 (397/648)	4/81	62/81	59.3 (96/162)
Hybrid2 ^[b]^	133/324	136/324	58.5 (379/648)	32/81	36/81	58.0 (94/162)
Milled Rice (Low: ≤0.2 ppm, High: ≥0.4 ppm)						
ALL varieties ^[a]^	120/405	143/405	67.5 (547/810 ^[N]^)	-	-	-
CL151 ^[b]^	139/324	144/324	56.3 (365/648)	62/81	1/81	61.1 (99/162)
Diamond ^[b]^	120/324	145/324	59.1 (383/648)	12/81	58/81	56.8 (92/162)
Hybrid1 ^[b]^	112/324	140/324	61.1 (396/648)	34/81	33/81	58.6 (95/162)
Gemini ^[b]^	101/324	119/324	66.0 (428/648)	24/81	47/81	56.2 (91/162)
Hybrid2 ^[b]^	119/324	129/324	61.7 (400/648)	32/81	40/81	55.6 (90/162)

^[a]^ Cross-validation model; ^[b]^ independent validation set; % CC = percent correct classification; ^[N]^, number of samples, sample size was based on the same 270 samples used in Table 3 and Table 7 but were repacked three times.

For the PLS analysis, across the five leave-one-variety-out models for rough rice, R^2^ for validation sets = 0.59 to 0.74 and with SEP = 2.24 to 2.85; brown rice had R^2^ = 0.74 to 0.78 and SEP = 1.02 to 1.12; and milled rice had R^2^ = 0.67 to 0.82 and SEP = 0.14 to 0.17 (Table 6). Discriminant analysis showed that the overall %CC for validation sets predicted from leave-one-variety-out calibration models had good potential for applications for rough rice, at %CC of 83.3 to 96.3, and milled rice, at 96.3 to 100%CC, while brown rice had the lowest range, at 66.0 to 77.8%CC. The lower performance for brown rice may be attributed to the higher complexity of the constituents in this rice form, especially when compared to milled rice, where bran has already been removed. Considering that Dors et al. [16] showed that the highest concentrations of pesticide residues were found in the bran fraction, it becomes even more critical that better prediction capabilities are targeted for brown rice. However, the larger market and supply of rice for consumption in the Philippines and many other countries remain to be milled rice. With the potential 96% to 100% correct classification for the detection of milled rice containing low versus high levels of CMPR, the use of a NIR multi-spectra LED instrument presents an additional viable technique that will readily assist in ensuring food safety for the rice industry. Although this study was also based on the absorption properties of chlorpyrifos-methyl in rough, brown, and milled rice, it is also good to try the method adopted by Yao et al. [13] in future studies, as it uses AVO-PLS and algorithms rather than PLS and simulations.

## 4. Conclusions

The potential for detecting the level or concentration of CMPR in rough, brown, and milled rice using a NIR LED-based instrument was shown. Based on the DA7200 analog, the averaged quantitative prediction models for the proposed LEDPrototype2 to predict CMPR in rough, brown, and milled rice had R^2^ = 0.64, 0.76, and 0.74, respectively. For qualitative prediction, the models showed an average percent correct classification of 89.98, 75.44, and 98.9 for rough, brown, and milled rice, respectively. These findings provided a concrete basis to pursue the design and development of a multi-spectral NIR Instrument (LEDPrototype2) that incorporates the use of the selected important wavelengths into the basic design concept of LEDPrototype1. The simulation process is limited to the wavelength specifications, angular emissions, and spikes for chlorpyrifos-methyl. Other noises and electrical components can also be taken into consideration for future studies. Likewise, the performance evaluation of the designed and developed multi-spectral NIR instrument will be pursued. These instruments will be highly beneficial to the rice industry, safety regulators, and consumers.

## Figures and Tables

**Figure 1 sensors-24-04055-f001:**
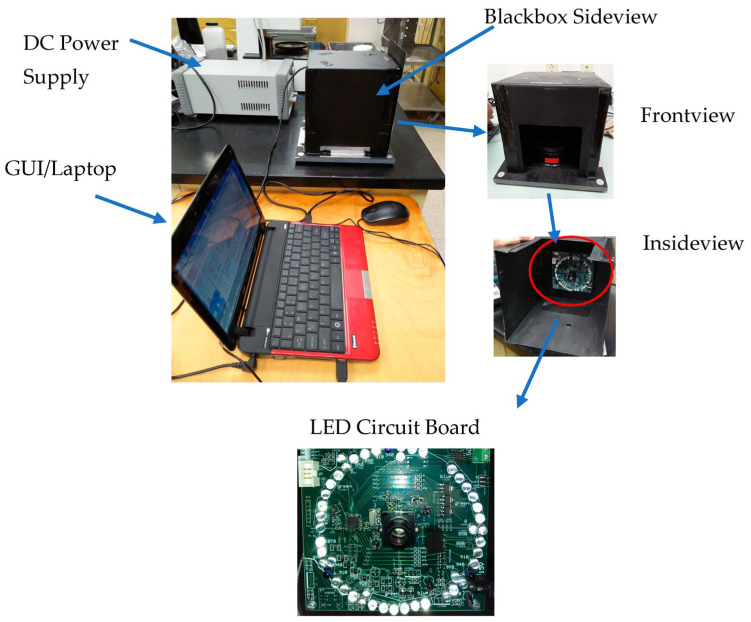
Photo of the vis/NIR LED-based instrument Prototype 1 showing the LED circuit board.

**Figure 2 sensors-24-04055-f002:**
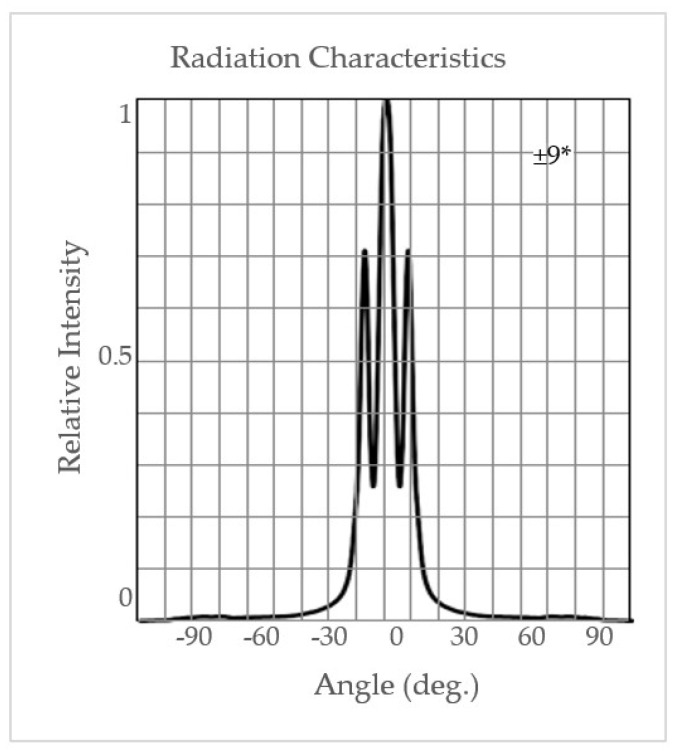
Angular emission characteristics for LEDs, L1300-6 are shown; * half-angle view at IF (forward current) = 50 mA degrees.

**Figure 3 sensors-24-04055-f003:**
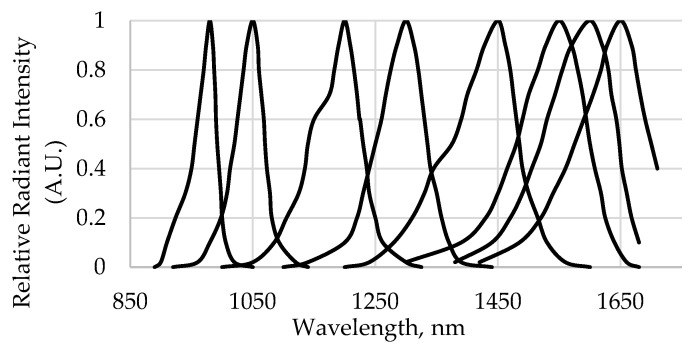
Plots showing overlapped relative spectral emissions of LEDs for wavelengths that were selected for the proposed NIR LED-based Prototype 2 instrument.

**Table 1 sensors-24-04055-t001:** Summary of significant wavelengths based on calibration models developed for rough, brown, and milled rice using Perten DA7200 and the list of discrete wavelengths fitted in the USDA-ARS vis/NIR LED-based Prototype 1 and recommendations for NIR LED-based Prototype 2.

LED Wavelength, nm	^[a]^ Marubeni LED	DA7200 Significant Prediction Wavelengths			USDA-ARS NIR LED-Based Prototype 1 (LEDPrototype1)	Proposed USDA-ARS NIR LED-Based Prototype 2 (LEDPrototype2)
		Rough Rice	Brown Rice	Milled Rice		
850					√	
910					√	
940					√	
970					√	
980	L980-06					√√
1050		X				√√
1070					√	
1200	L1200-06	X	X	X	√	√√
1300	L1300-06		X		√	√√
1360		X	X	X		*
1390		X				*
1410		X		X		*
1425		X				*
1450	L1450-06		X		√	√√
1470			X			*
1480		X				*
1510				X		*
1540		X		X		*
1550	L1550-06				√	√√
1580			X			*
1600	L1600-06					√√
1650	L1650-06					√√

X = important wavelengths based on calibrations; √ = LED wavelengths used in existing vis/NIR LED-based instrument; √√ = recommended wavelengths for proposed LED-based Protoype2 based on performance comparisons between Perten DA7200 and LED-based Prototype 1 and LED availability; ^[a]^ Marubeni America Corp., Santa Clara, CA, USA; * = not commercially available.

**Table 2 sensors-24-04055-t002:** PLS model prediction statistics for determination of chlorpyrifos-methyl residues based on the DA7200 analog for available wavelengths ^[a]^ in the LEDPrototype1 instrument.

Model Data	Calibration						Independent Validation		
	N	nF	R^2^ Cal	RMSEC	R^2^ CV	SECV	N	R^2^	SEP
Rough Rice (0 to 12 ppm)									
ALL varieties	270	8	0.68	2.39	0.60	2.66	-	-	-
CL151	216	8	0.70	2.31	0.63	2.58	54	0.56	2.91
Diamond	216	8	0.70	2.32	0.62	2.62	54	0.52	2.95
Hybrid1	216	8	0.66	2.47	0.57	2.79	54	0.71	2.28
Gemini	216	8	0.67	2.41	0.58	2.74	54	0.63	2.66
Hybrid2	216	8	0.69	2.35	0.60	2.68	54	0.62	2.65
Brown Rice (0 to 6 ppm)									
ALL varieties	269 *	7	0.66	1.24	0.63	1.30	-	-	-
CL151	216	7	0.67	1.22	0.63	1.29	53	0.63	1.31
Diamond	215	7	0.66	1.23	0.63	1.30	54	0.63	1.29
Hybrid1	215	7	0.67	1.22	0.64	1.28	54	0.58	1.38
Gemini	215	7	0.66	1.23	0.62	1.30	54	0.67	1.22
Hybrid2	215	7	0.65	1.26	0.61	1.33	54	0.70	1.17
Milled Rice (0 to 0.8 ppm)									
ALL varieties	270	8	0.73	0.15	0.69	0.16	-	-	-
CL151	216	8	0.76	0.14	0.71	0.15	54	0.58	0.19
Diamond	216	8	0.74	0.15	0.69	0.16	54	0.67	0.16
Hybrid1	216	8	0.74	0.15	0.69	0.16	54	0.68	0.16
Gemini	216	7	0.69	0.16	0.65	0.17	54	0.71	0.16
Hybrid2	216	6	0.68	0.16	0.64	0.17	54	0.70	0.16

^[a]^ Wavelengths at 80% relative radiant intensity include 1055–1085 nm, 1185–1215 nm, 1275–1320 nm, 1420–1470 nm, and 1515–1580 nm. N = number of samples; nF = number of factors used in PLS calibration model; R^2^ = coefficient of determination; RMSEC = root mean square error of calibration; Cal = calibration; CV = cross-validation; SECV = Standard error cross-validation; SEP = standard error of prediction. * The data from one sample are missing due to an error in saving the scanned sample using DA 7200. Nonetheless, it was well represented by 269 data points.

**Table 3 sensors-24-04055-t003:** Discriminant model prediction statistics for determination of chlorpyrifos-methyl residues based on the DA7200 analog for available wavelengths ^[a]^ in the LEDPrototype1 instrument.

Model Data	Calibration (Training Set)			Independent Validation (Test Set ^[b]^)		
	Number of False Positives	Number of False Negatives	Overall % CC	Number of False Positives	Number of False Negatives	Overall CC
Rough Rice (Low: ≤3.0 ppm, High: >3 ppm)						
ALL varieties	9/135	12/135	92.2 (249/270)	-	-	-
CL151	7/108	7/108	93.5 (202/216)	5/27	4/27	83.3 (45/54)
Diamond	6/108	8/108	93.5 (202/216)	7/27	5/27	77.8 (42/54)
Hybrid1	4/108	6/108	95.4 (206/216)	2/27	8/27	81.5 (44/54)
Gemini	7/108	10/108	92.1 (199/216)	4/27	1/27	90.7 (49/54)
Hybrid2	10/108	7/108	92.1 (199/216)	4/27	0/27	92.6 (50/54)
Brown Rice (Low: ≤1.5 ppm, High: >1.5 ppm)						
ALL varieties	30/134 ^[c]^	7/135	86.2 (232/269)	-	-	-
CL151	17/108	6/108	89.4 (193/216)	4/26	2/27	88.6 (47/53)
Diamond	20/107	6/108	87.9 (189/215)	7/27	4/27	79.6 (43/54)
Hybrid1	24/107	6/108	86.0 (185/215)	9/27	1/27	81.5 (44/54)
Gemini	23/107	6/108	86.5(186/215)	3/27	13/27	70.4 (38/54)
Hybrid2	20/107	5/108	88.4 (190/215)	9/27	1/27	81.5 (44/54)
Milled Rice (Low: ≤0.2 ppm, High: >0.2 ppm)						
ALL varieties	0/135	0/135	100.0 (270/270)	-	-	-
CL151	0/108	0/108	100.0 (216/216)	0/27	0/27	100.0 (54/54)
Diamond	0/108	0/108	100.0 (216/216)	5/27	0/27	90.7 (47/54)
Hybrid1	0/108	0/108	100.0 (216/216)	0/27	0/27	100.0 (54/54)
Gemini	0/108	0/108	100.0 (216/216)	0/27	0/27	100.0 (54/54)
Hybrid2	0/108	0/108	100.0 (216/216)	0/27	0/27	100.0 (54/54)

^[a]^ Wavelengths at 80% relative radiant intensity for Prototype 1 include 1055–1085 nm, 1185–1215 nm, 1275–1320 nm, 1420–1470 nm, and 1515–1580 nm. % CC = percent correct classification; ^[b]^ the test samples were those that were removed from the samples used to develop the calibration model; ^[c]^ one sample from brown rice is missing using DA 7200; thus, *n* = 269.

**Table 4 sensors-24-04055-t004:** PLS model prediction statistics for determination of chlorpyrifos-methyl residues using the actual LEDPrototype1 instrument.

Model Data	Calibration						Independent Validation		
	N	nF	R^2^ Cal	RMSEC	R^2^ CV	SECV	N	R^2^	SEP
Rough Rice (0 to 12 ppm)									
ALL varieties ^[a]^	810	4	0.43	3.20	0.42	3.23	-	-	-
CL151 ^[b]^	648	4	0.39	3.30	0.38	3.33	162	0.59	2.75
Diamond ^[b]^	648	5	0.42	3.23	0.40	3.27	162	0.51	2.98
Hybrid1 ^[b]^	648	5	0.41	3.23	0.40	3.28	162	0.53	2.92
Gemini ^[b]^	648	5	0.50	2.98	0.49	3.03	162	0.23	3.87
Hybrid2 ^[b]^	648	4	0.47	3.06	0.46	3.10	162	0.24	3.71
Brown Rice (0 to 6 ppm)									
ALL varieties ^[a]^	810	1	0.01	2.10	0.01	2.11	-	-	-
CL151 ^[b]^	648	5	0.10	2.00	0.07	2.04	162	0.03	2.10
Diamond ^[b]^	648	5	0.10	2.00	0.07	2.04	162	0.01	2.13
Hybrid1 ^[b]^	648	6	0.10	2.00	0.07	2.04	162	0.01	2.13
Gemini ^[b]^	648	1	0.01	2.10	0.00	2.11	162	0.03	2.11
Hybrid2 ^[b]^	648	1	0.01	2.10	0.01	2.11	162	0.00	2.11
Milled Rice (0 to 0.8 ppm)									
ALL varieties ^[a]^	810	3	0.05	0.27	0.04	0.28	-	-	-
CL151 ^[b]^	648	3	0.04	0.28	0.03	0.28	162	0.01	0.28
Diamond ^[b]^	648	3	0.06	0.27	0.04	0.28	162	0.01	0.29
Hybrid1 ^[b]^	648	3	0.08	0.27	0.07	0.27	162	0.01	0.29
Gemini ^[b]^	648	3	0.08	0.27	0.06	0.27	162	0.01	0.28
Hybrid2 ^[b]^	648	1	0.00	0.28	0.00	0.28	162	0.02	0.28

^[a]^ Cross-validation model; ^[b]^ independent validation set; N = number of samples, sample size was based on the same 270 samples but repacks 3 times; nF = number of factors used in PLS calibration model; R^2^ = coefficient of determination; RMSEC = root mean square error of calibration; Cal = calibration; CV = cross-validation; SECV = standard error cross-validation; SEP = standard error of prediction.

**Table 6 sensors-24-04055-t006:** PLS model prediction statistics for determination of chlorpyrifos-methyl residues using the DA7200 analog for selected wavelengths ^[a]^ of the NIR LED-based Prototype 2 instrument.

Model Data	Calibration						Independent Validation		
	N	nF	R^2^ Cal	RMSEC	R^2^ CV	SECV	N	R^2^	SEP
Rough Rice (0 to 12 ppm)									
ALL RR varieties	270	10	0.74	2.2	0.63	2.6	-	-	-
CL151	216	9	0.72	2.2	0.64	2.5	54	0.59	2.85
Diamond	216	10	0.75	2.1	0.64	2.5	54	0.63	2.62
Hybrid1	216	10	0.73	2.2	0.63	2.6	54	0.74	2.24
Gemini	216	10	0.76	2.1	0.65	2.5	54	0.64	2.71
Hybrid2	216	10	0.76	2.1	0.66	2.5	54	0.61	2.70
Brown Rice (0 to 6 ppm)									
ALL BR varieties	269 *	8	0.78	0.99	0.75	1.07	-	-	-
CL151	216	8	0.78	0.99	0.74	1.07	53	0.74	1.12
Diamond	215	8	0.77	1.01	0.74	1.08	54	0.77	1.03
Hybrid1	215	8	0.77	1.01	0.74	1.08	54	0.78	1.02
Gemini	215	8	0.79	0.98	0.76	1.05	54	0.75	1.07
Hybrid2	215	8	0.78	1.00	0.74	1.08	54	0.78	1.02
Milled Rice (0 to 0.8 ppm)									
ALL MR varieties	270	8	0.76	0.14	0.73	0.15	-	-	-
CL151	216	8	0.78	0.13	0.74	0.14	54	0.67	0.17
Diamond	216	8	0.76	0.14	0.72	0.15	54	0.75	0.14
Hybrid1	216	8	0.78	0.13	0.73	0.15	54	0.74	0.14
Gemini	216	8	0.76	0.14	0.71	0.15	54	0.82	0.13
Hybrid2	216	8	0.78	0.13	0.73	0.15	54	0.71	0.16

^[a]^ Wavelengths at 80% relative radiant intensity for Prototype 2 include 970–985 nm, 1035–1060 nm, 1185–1215 nm, 1275–1320, 1420–1470 nm, 1515–1580 nm, 1555–1630, and 1620–1680 nm; N = number of samples; F = PLS factor number; nF = number of factors used in PLS calibration model; R^2^ = coefficient of determination; RMSEC = root mean square error of calibration; Cal = calibration; CV = cross-validation; SECV = standard error cross-validation; SEP = standard error of prediction. * Data from one sample are missing due to an error in saving the scanned sample using DA 7200. Nonetheless, it was well represented by 269 data points.

**Table 7 sensors-24-04055-t007:** Discriminant model prediction statistics for determination of chlorpyrifos-methyl residues based on the DA7200 analog for the selected wavelengths ^[a]^ of the NIR LED-based Prototype 2 instrument.

Model Data	Calibration (Training Set)			Independent Validation (Test Set ^[b]^)		
	Number of False Positives	Number of False Negatives	Overall % CC	Number of False Positives	Number of False Negatives	Overall % CC
Rough Rice (Low: ≤3.0 ppm, High: >3.0 ppm)						
ALLQualRR	5/135	8/135	95.2 (257/270)	-	-	-
CL151	3/108	6/108	95.8 (207/216)	2/27	3/27	90.7 (49/54)
Diamond	0/108	5/108	97.7 (211/216)	4/27	5/27	83.3 (45/54)
Hybrid1	1/108	4/108	97.7 (211/216)	0/27	4/27	92.6 (50/54)
Gemini	2/108	7/108	95.8 (207/216)	5/27	2/27	87.0 (47/54)
Hybrid2	6/108	4/108	95.4 (206/216)	1/27	1/27	96.3 (52/54)
Brown Rice (Low: ≤1.5 ppm, High: >1.5 ppm)						
ALLQualBR	27/134 ^[c]^	4/135	88.5 (238/269)	-	-	-
CL151	13/108	5/108	91.7 (198/216)	13/26	5/27	66.0 (35/53)
Diamond	19/107	1/108	90.7 (195/215)	6/27	6/27	77.8 (42/54)
Hybrid1	19/107	4/108	89.3 (192/215)	9/27	3/27	77.8 (42/54)
Gemini	17/107	2/108	91.2 (196/215)	10/27	2/27	77.8 (42/54)
Hybrid2	17/107	4/108	90.2 (194/215)	8/27	4/27	77.8 (42/54)
Milled Rice (Low: ≤0.2 ppm, High: >0.2 ppm)						
ALLQualMR	0/135	0/135	100.0 (270/270)	-	-	-
CL151	0/108	0/108	100.0 (216/216)	0/27	0/27	100.0 (54/54)
Diamond	0/108	0/108	100.0 (216/216)	1/27	1/27	96.3 (52/54)
Hybrid1	0/108	0/108	100.0 (216/216)	0/27	1/27	98.2 (53/54)
Gemini	0/108	0/108	100.0 (216/216)	0/27	0/27	100.0 (54/54)
Hybrid2	0/108	0/108	100.0 (216/216)	0/27	0/27	100.0 (54/54)

^[a]^ Wavelengths at 80% relative radiant intensity for Prototype 2 include 970–985 nm, 1035–1060 nm, 1185–1215 nm, 1275–1320 nm, 1420–1470 nm, 1515–1580 nm, 1555–1630 nm, and 1620–1680 nm; % CC = percent correct classification; ^[b]^ the test samples were those that were removed from the samples used to develop the calibration model; ^[c]^ one sample from brown rice is missing using DA 7200, thus *n* = 269.

## Data Availability

Data are contained within the article.

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
