# Peer review of "Developing a Multi-Spectral NIR LED-Based Instrument for the Detection of Pesticide Residues Containing Chlorpyrifos-Methyl in Rough, Brown, and Milled Rice"

_sensors, 2024, doi:10.3390/s24134055_

Round 1

Reviewer 1 Report

Comments and Suggestions for Authors

1.     Lines 242 through 244 of the explanation of Table 2 contain inconsistencies in the formatting of the explanation of symbols, with the absence of an equal sign after R² and SECV.

2.     This article has samples of multiple species with multiple concentration gradients, please explain in detail the specific division of the validation set.

3.     In the 117-120 sentences, three different kinds of rice use different pesticide concentrations, may I ask what is the theoretical basis or experimental basis for this?

4.     In sentences 169-172, does the use of soap and water to clean the sample tray have any effect on the detection, or is there a theoretical basis and experiment to prove that doing so has no effect on the detection?

5.     In sentences 181 to 183, please explain the reason for not conducting independent verification.

6.     Line 217. The objective proposed in this study is the detection of CMPR, but the recommended wavelengths provided in Table 1 are significantly different from the wavelengths predicted to be important based on the DA7200 (with only the wavelengths 1050, 1200 for Rough Rice; 1200, 1300, 1450 for Brown Rice; and 1200 for Milled Rice being the same). There is a need to provide compelling reasons for the recommended wavelengths.

7.     Line 227. In the simulation with DA7200 analogue, the selected bands (1055-1085 nm, 1185-1215 nm, 1275-1320 nm, 1420-1470 nm, and 1515-1580 nm) are composed of many variables, which is different from the principle of the LED-based Prototype. In fact, an LED can only provide a single output intensity value, even though its emission range covers the selected bands.

Comments on the Quality of English Language

No Comments

Reviewer 2 Report

Comments and Suggestions for Authors

The manuscript deals with an interesting and actual topic which could be of interest to the audience of Sensors.

However, in my opinion, the paper suffers for important methodological issues and lack of clarity deserving major improvements before being reconsidered for publication.

Following there are the major concerns.

Materials and methods

- lines 99-124: the sampling plan is not clear and deserves huge improvement in clarity. For example, is not clear how the total number of samples became 90; which is the number of independent samples; if milling samples are obtained only from brown rice and why.

- lines 132-135: please furnish others info such as the resolution, the number of scans per sample, the management of the background.

- lines 174-187: is it correct that a total of 270 spectra per each rice type were acquired? How is it possible? Probably, there are technical replicates into the dataset. Please specify. And, linked to this aspect, if there are technical replicate, it is a big issue if those have been split into calibration and validation sets without taking into account that the same sample (i.e., the replicates of the same sample) could not be part of both the calibration and validation sets. Otherwise, it is not a real and genuine validation set. All these aspects should be clearly reported.

Results and discussion

- the simulation referred as DA7200 analogue should be presented in material and methods, together with all the technical aspects that are briefly reported in the results section. Moreover, the rationale for this trial should be clarified.

- the number of the samples (i.e., N) changes in the tables. This add more confusion in understanding exactly the experimental plan that should be well and clearly reported in materials and methods.

- no info are reported about the development and the features (e.g., number of LVs) of the PLS-DA models.

- more in general, the poor performance observed in predicting the amount of the pesticide could be associated to the unfeasibility of NIR spectroscopy to measure it? Because of the small concentrations or for other reasons? This should be taken into account and critically discussed in my opinion.

Comments on the Quality of English Language

The language could be improved to make the paper more clear.

Reviewer 3 Report

Comments and Suggestions for Authors

The paper concerns the DA Perten 7200 NIR Spectrometer detecting chlorpyrifos-methyl pesticide residue in rough, brown, and milled rice. The manuscript is acceptable in terms of technical aspects. However, the manuscript requires additional work and major revision before publication in the journal. To improve the manuscript, I recommend the authors address the following points.

1. Abstract. The sentence should be rewritten. “An actual LED based instrument with this capability could provide a quick screening tool to determine if MRLs are exceeded.”

2. Keywords. “Detection” should be deleted. “LED” and “NIR” should not be abbreviated.

3. Introduction. 1) Line 80: "...by Rodriguez et al. (2020) using..." should cite reference. 2) Some sentences should cite references, such as lines 49-83. 3) The aim of the manuscript should be explained better in the introduction.

4. Materials and Methods. 1) Line 100: "used by Rodriguez et al. (2020)." should cite the reference. 2) Figures 1 and 2 are not clear. Please revise them.

5. "3. Results" should be revised as "3. Results and Discussion".

6. Results and Discussion. Authors are advised to discuss and, if possible, compare their results with those of other similar studies and add corresponding references.

7. Conclusion. 1) "5. Conclusions" should be revised as "4. Conclusions ". 2) This part contains too much data. 3) Discuss the potential limitations and challenges of the proposed method.

8. References: Authors should cite the literature from the last five years. Ensure that all relevant references are up-to-date and accurately cited.

Comments on the Quality of English Language

Minor editing of English language required

Round 2

Reviewer 2 Report

Comments and Suggestions for Authors

Although the authors tried to address the points previously emerged, in my opinion still following the rationale of the work in particular regarding how the data are treated by the multivariate methods is exstremely difficult.

Again, no clear and direct explanation of why the number in the tables changed are funished and, as a results, a reader should read the paper together with his calculator to understand the sampling scheme;

As far as I can understand, the information required for the discriminant methods (e.g., the number of LVs) have not been reported;

More in general, in my opinion the paper presents still serious lacks, mostly in terms of clarity, that strongly and negatively influence the quality of the paper.

Comments on the Quality of English Language

Could be improved.

Author Response

Response:  Thank you for pointing this out.  We agree with this comment.  Therefore, we made necessary changes in Tables 2, 4, and 6 by changing F to nF or the number of factors used in PLS calibration model, and in Tables 3, 5, and 7 by adding the LVs (shown in Overall % CC the division of samples), likewise the insertion of other terms, for calibration as a training set, and Independent validation as a test set. 

Reviewer 3 Report

Comments and Suggestions for Authors

This manuscript was revised well and can be published as it is.

Comments on the Quality of English Language

Minor revision is requested.

Author Response

Thank you very much for working on this paper. We edited the English language by an English major professor and the use of Grammarly application.